# Using Deauville Scoring to Guide Consolidative Radiotherapy in Diffuse Large B-Cell Lymphoma

**DOI:** 10.3390/cancers16193311

**Published:** 2024-09-27

**Authors:** Chun En Yau, Chen Ee Low, Whee Sze Ong, Lay Poh Khoo, Joshua Tian Ming Hoe, Ya Hwee Tan, Esther Wei Yin Chang, Valerie Shiwen Yang, Eileen Yi Ling Poon, Jason Yongsheng Chan, Iris Huili Sin, Kheng Wei Yeoh, Nagavalli Somasundaram, Mohamed Farid Bin Harunal Rashid, Miriam Tao, Soon Thye Lim, Jianbang Chiang

**Affiliations:** 1Yong Loo Lin School of Medicine, National University of Singapore, 10 Medical Dr, Singapore 117597, Singapore; e0886363@u.nus.edu (C.E.Y.); cheneelow@u.nus.edu (C.E.L.); 2Division of Clinical Trials and Epidemiological Sciences, National Cancer Centre Singapore, 11 Hospital Crescent, Singapore 169610, Singapore; ong.whee.sze@singhealth.com.sg; 3Division of Medical Oncology, National Cancer Centre Singapore, 11 Hospital Crescent, Singapore 169610, Singapore; khoo.lay.poh@nccs.com.sg (L.P.K.); joshua.hoe.t.m@singhealth.com.sg (J.T.M.H.); tan.ya.hwee@singhealth.com.sg (Y.H.T.); esther.chang.w.y@singhealth.com.sg (E.W.Y.C.); valerie.yang.s.w@singhealth.com.sg (V.S.Y.); eileen.poon.y.l@singhealth.com.sg (E.Y.L.P.); jason.chan.y.s@singhealth.com.sg (J.Y.C.); nagavalli.somasundaram@singhealth.com.sg (N.S.); mohamad.farid@singhealth.com.sg (M.F.B.H.R.); miriam.tao@singhealth.com.sg (M.T.); lim.soon.thye@singhealth.com.sg (S.T.L.); 4Duke-NUS Medical School, Oncology Academic Clinical Program, 8 College Road, Singapore 169857, Singapore; 5Division of Radiation Oncology, National Cancer Centre Singapore, 11 Hospital Crescent, Singapore 169610, Singapore; iris.sin.huili@singhealth.com.sg (I.H.S.); yeoh.kheng.wei@singhealth.com.sg (K.W.Y.)

**Keywords:** PET-CT, DLBCL, consolidative treatment, de-escalation, radiotherapy

## Abstract

**Simple Summary:**

This study aims to evaluate the role of end-of-treatment PET-CT scans, interpreted using the Deauville score (DV), in guiding the use of consolidative radiotherapy (RT) for DLBCL patients. The goal is to help avoid unnecessary RT for low-risk patients, as current guidelines are unclear, potentially leading to the overuse of RT. We analyzed the data of 349 patients and RT was associated with a significant improvement in time-to-progression amongst the DV4-5 patients but not the DV1-3 patients. Our data suggest that DLBCL patients with end-of-treatment PET-CT DV1-3 may not require consolidative RT.

**Abstract:**

Background: The most common aggressive lymphoma in adults is diffuse large B-cell lymphoma (DLBCL). Consolidative radiotherapy (RT) is often administered to DLBCL patients but guidelines remain unclear, which could lead to unnecessary RT. We aimed to evaluate the value of end-of-treatment PET-CT scans, interpreted using the Deauville score (DV), to guide the utilization of consolidative RT, which may help spare low-risk DLBCL patients from unnecessary RT. Methods: We included all DLBCL patients diagnosed between 2010 and 2022 at the National Cancer Centre Singapore with DV measured at the end of the first-line chemoimmunotherapy. The outcome measure was time-to-progression (TTP). The predictive value of DV for RT was assessed based on the interaction effect between the receipt of RT and DV in Cox regression models. Results: The data of 349 patients were analyzed. The median follow-up time was 38.1 months (interquartile range 34.0–42.3 months). RT was associated with a significant improvement in TTP amongst the DV4-5 patients (HR 0.33; 95%CI 0.13–0.88; *p* = 0.027) but not the DV1-3 patients (HR 0.85; 95%CI 0.40–1.81; *p* = 0.671) (interaction’s *p* = 0.133). Multivariable analysis reported that RT was again significantly associated with improved TTP among the DV4-5 patients (adjusted HR 0.29; 95%CI 0.10–0.80; *p* = 0.017) but not the DV1-3 group (HR 0.86; 95%CI 0.40–1.86; *p* = 0.707) (interaction’s *p* = 0.087). Conclusion: Our results suggests that DLBCL patients with end-of-treatment PET-CT DV1-3 may not need consolidative RT. Longer follow-up and prospective randomized trials are still necessary to investigate long-term outcomes.

## 1. Introduction

Since 2011, diffuse large B-cell lymphoma (DLBCL) annual mortality has increased over 3.5% annually, ranking it as the most common aggressive lymphoma in adults [1,2]. Despite recent treatment advances achieving a 5-year survival of 60% to 70%, up to 50% of DLBCL patients eventually relapse [3,4]. Currently, six cycles of rituximab with cyclophosphamide, doxorubicin, vincristine, and prednisolone (R-CHOP) remain the standard of care for most patients with advanced disease [5,6]. International guidelines recommend using PET-CT with 18F-fluorodeoet xyglucose (18F-FDG PET-CT) to stage and assess remission in DLBCL patients [7]. There are various meta-analyses with supporting evidence on the prognostic value of 18F-FDG PET-CT used at the end of treatment [8,9,10].

To standardize the interpretation of PET-CT scans, the Deauville five-point scale (DV) is commonly used [11,12]. It provides an objective measure for the assessment of the metabolic response by comparing the PET-CT avidity of focal abnormalities to liver and mediastinal uptake [13,14]. The DV scoring system is recommended by current guidelines for the PET-CT-based evaluation of treatment response [15]. Studies have shown that DV applied prior to autologous stem cell transplantation could be a powerful determinant of prognosis after autologous stem cell transplant (ASCT) [16,17]. 

Based on existing guidelines, consolidative radiation therapy (RT) is used after the completion of standard treatment for those with residual masses or bulky disease [18]. However, there has not been a randomized controlled trial of consolidative RT performed in the contemporary era with the use of multiagent chemoimmunotherapy incorporating rituximab [19]. Clinicians have to weigh up between the toxicities of RT, which can vary widely in symptoms and severity with the tissue being irradiated [20], and the toxicities of other salvage treatments, seeing that the response to the primary treatment of DLBCL is highly predictable of long-term outcome.

As a result, it is of clinical interest to refine patient selection to prevent patients from receiving unnecessary toxic consolidative treatment. Increasingly, studies are incorporating a PET-CT scan at the end of standard treatment to select patients who may benefit from consolidative RT. This approach could spare low-risk patients from potentially unwarranted RT and selectively administer RT only to those deemed at high risk. Freeman et al. recently published the results of their end-of-treatment PET-CT-guided approach to select patients for RT [21]. Out of 723 DLBCL patients, they found that patients with advanced DLBCL who have a negative end-of-treatment PET-CT scan have an excellent prognosis without RT. Despite the omission of RT, PET-CT-negative patients with bulky disease had similar outcomes from those without bulk. Their data suggest that 18F-FDG PET-CT can effectively guide the selective administration of RT. 

In this study, we aimed to assess the value of end-of-treatment PET-CT scans, interpreted using the Deauville score (DV), in guiding the utilization of consolidative RT. This approach may help spare low-risk DLBCL patients from unnecessary RT. We report on the utility of using the DV score at end-of-treatment PET-CT scans to guide the administration of consolidative RT in patients with DLBCL.

## 2. Methods

### 2.1. Study Cohort and Design

From a prospectively maintained database of patients diagnosed and treated for DLBCL at the National Cancer Centre Singapore, we identified all DLBCL patients diagnosed from 2010 to 2022 with DV measured at the end of first-line chemoimmunotherapy. Patients who received treatment apart from consolidative RT after their first-line chemoimmunotherapy were excluded. Selected patients were separated into two groups based on whether they received consolidative RT (RT-treated vs. RT-omitted) for comparison. The decision for patients to receive RT was made by a multidisciplinary tumor board. The research study was carried out as part of the Singapore Lymphoma Study with approval from the SingHealth Centralised Institutional Review Board (CIRB 2018/3084). This study conforms to the Declaration of Helsinki. 

### 2.2. Variables and Outcome Measure

Extracted information included patient demographics (age, race, and sex), clinical characteristics (presence of B-symptoms, Eastern Cooperative Oncology Group [ECOG] performance status, Ann Arbor staging, International Prognostic Index [IPI], bone marrow involvement, number of extranodal sites, and bulky status of tumors), and biochemical data such as serum lactate dehydrogenase (LDH) and hemoglobin (Hb) levels at diagnosis before treatment. Deauville scores were derived from end-of-treatment PET-CT scans, performed in accordance with the Lugano Criteria. These scans are routinely performed 6–8 weeks upon completion of chemoimmunotherapy treatment, and radiotherapy is administered expediently within 4 weeks as per institutional practice in our center.

The outcome measure was time-to-progression (TTP), defined as the time from diagnosis to progression or relapse of DLBCL, or death due to DLBCL or treatment-related factors. Patients without these events were censored at their last follow-up. 

### 2.3. Statistical Analysis

Comparisons of continuous variables between RT-treated and RT-omitted patients were performed based on t-test. Corresponding comparisons of categorical variables were based on either the Pearson’s Chi-squared test or Fisher’s exact test, as appropriate. Follow-up duration was estimated using the reverse Kaplan–Meier method. TTP was estimated using the Kaplan–Meier method. The association of TTP with each characteristic was assessed via the Cox proportional hazard model and tested using Wald’s test. The predictive value of the DV risk group (DV1-3 vs. DV4-5) for RT was analyzed by including an interaction term between receipt of RT and DV risk group in a Cox model. Variables with a *p*-value ≤ 0.1 in their univariate Cox model and those deemed clinically significant were included in multivariable Cox analyses. All statistical tests were 2-sided and a *p*-value < 0.05 was considered statistically significant. All statistical analyses were performed using R software (version 4.1.2).

## 3. Results 

### 3.1. Patient Characteristics

From 2010 to 2022, 349 eligible patients were included (Appendix A). In total, 80 of the included patients were treated with RT (RT-treated), with 13 of them having DV4-5 at the end of first-line chemoimmunotherapy. Meanwhile, 269 patients were treated without RT (RT-omitted), with 30 of them having DV4-5 at the end of first-line chemoimmunotherapy. The patient characteristics are detailed in Table 1. All characteristics were largely similar between both groups, except that proportionally more RT-treated patients had stage 1–2 disease (61% vs. 43%), elevated lactate dehydrogenase (85% vs. 71%), and bulky disease (25% vs. 10%) compared to RT-omitted patients. On average, RT-treated patients also had lower baseline hemoglobin levels compared to RT-omitted patients (11.7 g/dL vs. 12.2 g/dL, *p* = 0.047).

### 3.2. Prognostic Factors for TTP

The median follow-up time of the entire cohort was 38.1 months (interquartile range 34.0–42.3 months). Univariable and multivariable Cox analysis results are presented in Table 2 and Table 3, respectively. Receipt of consolidative RT, DV, and presence of B symptoms were found to be significantly associated with TTP.

### 3.3. Predictive Value of DV for RT

The Kaplan Meier curve is shown in Figure 1. The TTP HR was 0.81 (95%CI 0.45–1.48; *p* = 0.492) for a 19% reduction in progression with RT versus RT-omitted patients (Table 4). The estimated 2-year TTP was 82% (95%CI 73–93%) for the RT-treated group versus 82% (95%CI 78–87%) for the RT-omitted group.

In the DV4-5 subgroup, the TTP HR was 0.33 (95%CI 0.13–0.88; *p* = 0.027) for a 67% reduction in progression with RT versus RT-omitted patients (Table 4). Within this subgroup, the estimated 2-year TTP was 57% (95%CI 34–95%) for the RT-treated group versus 32% (95%CI 19–55%) for the RT-omitted group.

In the DV1-3 subgroup, the TTP HR of the RT-treated patients was 0.85 (95%CI 0.40–1.81; *p* = 0.671) when compared with RT-omitted patients (Table 4). Within this subgroup, the estimated 2-year TTP was 87% (95%CI 77–97%) for the RT-treated group versus 89% (95%CI 84–93%) for the RT-omitted group. The *p*-value of the interaction between receipt of RT and DV status was 0.133. 

Adjusting the subgroup analysis for International Prognostic Index, presence of B symptoms, bone marrow involvement, and hemoglobin levels on multivariable analysis, the TTP outcomes by receipt of consolidative RT were different depending on the DV risk group of the patients. In the DV4-5 subgroup, the TTP-adjusted HR was 0.29 (95%CI 0.10–0.80; *p* = 0.017) for a 71% reduction in progression with RT versus RT-omitted patients (Table 4). In the DV1-3 subgroup, the TTP-adjusted HR of the RT-treated patients was 0.86 (95%CI 0.40–1.86; *p* = 0.707) when compared with the RT-omitted patients (Table 4). Within this adjusted subgroup analysis, the *p*-value of the interaction between receipt of RT and DV status was 0.087.

### 3.4. Sensitivity Analysis

As a higher proportion of RT-treated patients was diagnosed later than RT-omitted patients, median follow-up times were significantly different between the two groups (RT-treated: 25.4 months vs. RT-omitted: 42.3 months, *p* < 0.001). To examine the impact of the different follow-up durations between the RT-treated and RT-omitted patients on the subgroup analysis, we performed a sensitivity analysis censoring TTP at 24 months (Table 5). 

In the DV4-5 subgroup of this sensitivity analysis, the TTP-adjusted HR was 0.28 (95%CI 0.10–0.79; *p* = 0.017) for a 72% reduction in progression with RT versus RT-omitted patients (Table 5). In the DV1-3 subgroup, the TTP HR of the RT-treated patients was 0.84 (95%CI 0.34–2.08; *p* = 0.711) when compared with the RT-omitted patients (Table 5). The *p*-value of the interaction between receipt of RT and DV status was 0.114.

We also performed a sensitivity analysis examining the subset of patients who received at least six cycles of chemoimmunotherapy and did not progress before the end of the chemoimmunotherapy regimen. This reduces the bias that could be incurred by including patients who progressed while on the original chemoimmunotherapy regimen. The Kaplan–Meier curves of this subgroup of patients are in Appendix A. We then performed Cox regression on this subset of patients. The multivariable survival analysis and interaction analysis of this subgroup are in Appendix A. The results do not differ significantly from our main analyses, where consolidative RT was not significantly associated with the TTP of DLBCL patients with DV1-3 scores, with the *p*-value of the interaction between receipt of RT and DV status being 0.321.

As an exploratory analysis, we performed further sensitivity analysis to determine if consolidative RT affects the survival of patients with DV4 and DV5 differently. The Kaplan–Meier curve of this analysis is in Appendix A. A log-rank test comparing the DV4 and DV5 patients indicated a significant difference in TTP between the two groups of patients (*p* < 0.001), and another log-rank test comparing the four groups of patients (DV4 + No RT, DV5 + No RT, DV4 + RT, and DV5 + RT) indicated that there is a significant difference in the TTP of these four groups of patients (*p* < 0.001). However, we recommend the interpretation of these results with caution due to the small numbers.

## 4. Discussion

Guidelines concerning the efficacy of RT in the treatment of DLBCL are still unclear due to the presence of conflicting evidence [22]. Opting for consolidative RT in suitable patients is considered a viable and better option when compared to salvage chemoimmunotherapy and consolidative autologous stem cell transplant [21]. However, unnecessary RT can potentially cause unwanted morbidity in DLBCL patients. Given the lack of randomized clinical trials demonstrating the efficacy of consolidative RT in the contemporary era, evidence regarding the clinical outcomes of consolidative RT in DLBCL patients is much needed. In this study, we assessed the potential of utilizing DV scoring as a guide to treat patients with consolidative RT. We have demonstrated that consolidative RT was not significantly associated with the TTP of DLBCL patients with DV1-3 scores at the end of first-line chemoimmunotherapy (adjusted HR = 0.86; 95% CI 0.40–1.86) as compared with those patients with DV4-5 scores (adjusted HR = 0.29; 95% CI 0.10–0.80). Barring a randomized controlled trial (RCT) directly comparing patients treated either with consolidative RT or a placebo, we cannot conclude that consolidative RT should be excluded in DV1-3 patients based on the analyses from our retrospective study. However, our study offers real-world evidence of the possible efficacy of this approach and should be taken into consideration by practicing clinicians and in the future planning of trials.

Existing studies [9,23,24,25] have proposed DV scoring as a prognostic variable for progression-free survival and overall survival (OS), with some studies [26] even suggesting that metabolic responses determined by DV scoring can be explored as surrogate endpoints. In the GOYA trial [26], where patients were randomized to obinutuzumab plus cyclophosphamide, doxorubicin, vincristine, and prednisone (G-CHOP) vs. standard-of-care rituximab plus CHOP, all patients received a PET-CT scan at baseline and at the end of treatment. This study demonstrated that complete response as determined by end-of-treatment PET-CT results interpreted using the Cheson 2007 [27] and standard Lugano 2014 [28] response criteria was highly prognostic for PFS and OS. On the other hand, Pregno et al. [23] found that interim PET-CT scans failed to prognosticate outcomes of DLBCL patients. Another study [11] on the DV scoring of 212 Hodgkin lymphoma patients showed that patients with DV-1 to 3 were continued on the same treatment plan with good outcomes. Future observational studies and trials will require a longer follow-up duration to ascertain the use of DV as a potential marker to guide DLBCL patient selection for consolidative RT. More research is also needed to establish the balance between the accuracy and timeliness of prognostication. It is also interesting to note that patients with PET-NEG residual masses have higher relapse risks and highlights a subgroup of patients who may benefit from RT post chemotherapy [29]. 

There are several limitations in our results. Firstly, we do not have adequate follow-up duration to assess OS in our patients. In this study, we used TTP as a surrogate for OS, as patients who have poor survival prognosis tend to relapse within 24 months [30]. We also attempted sensitivity analysis to assess the impact of unequal follow-up duration between RT-treated and RT-omitted patients on TTP results, and we found that the different follow-up duration made no distinct changes to our study conclusion. Secondly, we had relatively few patients who had DV3 status, rendering it untenable to analyze this group by itself. Moving forward, we plan to continue to follow up on these patients to analyze their OS outcome. Thirdly, more than two-thirds of patients received the R-CHOP regimen, among whom ≥95% received ≥6 cycles, and 94.3% of patients across all regimens received at least 6 cycles of the treatment. However, there could be residual heterogeneity due to the different regimens and doses administered in the remaining patients. This reflects real-world practice where patients may not be fit upfront for multiagent doxorubicin-based chemoimmunotherapy. Future RCTs should consider randomizing patients based on DV status to observe the benefits of consolidative RT. Fourthly, our current dataset does not capture the dose reduction between the RT and non-RT patients. Future research should prospectively track the dose reductions and adjust for its effect on survival outcomes.

## 5. Conclusions

Our study suggests that DLBCL patients who received first-line chemoimmunotherapy with end-of-treatment PET-CT DV1-3 may not require consolidative RT. This observation will need confirmation with an RCT to ascertain the use of DV as a potential marker to guide patient selection for consolidative RT. 

## Figures and Tables

**Figure 1 cancers-16-03311-f001:**
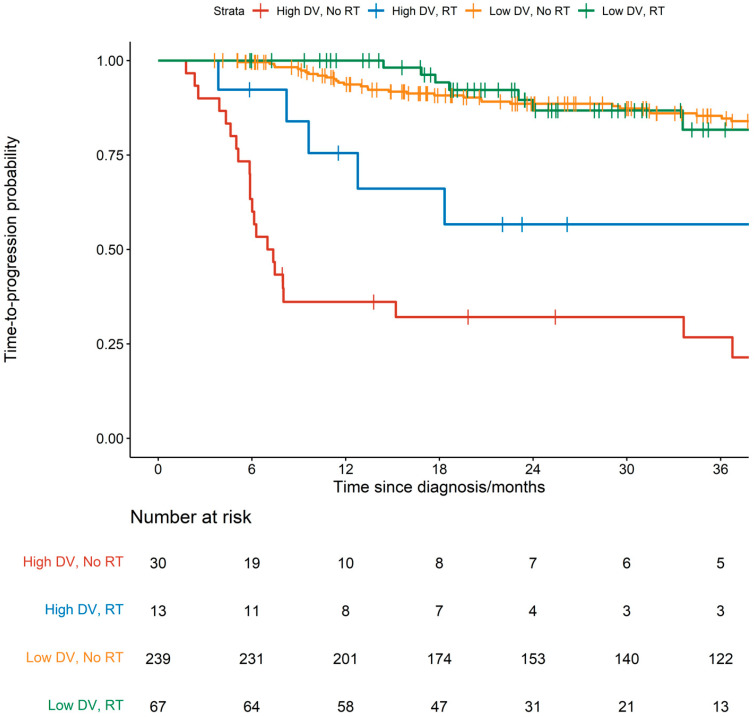
Combined graph by Deauville and radiotherapy status.

**Table 1 cancers-16-03311-t001:** Patient characteristics.

Characteristic	RT-Omitted, N = 269	RT-Treated, N = 80	*p*-Value ^2^
Gender			0.616
Female	133/269 (49%)	37/80 (46%)	
Male	136/269 (51%)	43/80 (54%)	
Age at Diagnosis ^1^	60.18 (15.18)	58.33 (18.03)	0.404
Deauville Score			0.008
1	145/269 (54%)	30/80 (38%)	
2	64/269 (24%)	21/80 (26%)	
3	30/269 (11%)	16/80 (20%)	
4	12/269 (4.5%)	10/80 (13%)	
5	18/269 (6.7%)	3/80 (3.8%)	
Deauville Score			0.223
DV4-5	30/269 (11%)	13/80 (16%)	
DV1-3	239/269 (89%)	67/80 (84%)	
IPI Risk Group			0.151
1	82/269 (30%)	33/80 (41%)	
2	77/269 (29%)	20/80 (25%)	
3	66/269 (25%)	13/80 (16%)	
4	37/269 (14%)	14/80 (18%)	
Unavailable	7/269 (2.6%)	0/80 (0%)	
Stratified IPI Risk Group			0.276
High ≥ 3	103/269 (38%)	27/80 (34%)	
Low < 3	159/269 (59%)	53/80 (66%)	
Unavailable	7/269 (2.6%)	0/80 (0%)	
Bulky Status			0.003
Bulky	28/269 (10%)	20/80 (25%)	
Not Bulky	216/269 (80%)	52/80 (65%)	
Unavailable	25/269 (9.3%)	8/80 (10%)	
ECOG Status			0.571
0	162/269 (60%)	53/80 (66%)	
1	89/269 (33%)	24/80 (30%)	
2	8/269 (3.0%)	3/80 (3.8%)	
3	3/269 (1.1%)	0/80 (0%)	
Unavailable	7/269 (2.6%)	0/80 (0%)	
Stratified ECOG Status			0.526
High ≥ 2	11/269 (4.1%)	3/80 (3.8%)	
Low < 2	251/269 (93%)	77/80 (96%)	
Unavailable	7/269 (2.6%)	0/80 (0%)	
B Symptoms			0.117
No	150/269 (56%)	35/80 (44%)	
Yes	74/269 (28%)	31/80 (39%)	
Unavailable	45/269 (17%)	14/80 (18%)	
Elevated LDH			0.039
No	72/269 (27%)	11/80 (14%)	
Yes	192/269 (71%)	68/80 (85%)	
Unavailable	5/269 (1.9%)	1/80 (1.3%)	
Extranodal Involvement			0.105
No	109/269 (41%)	23/80 (29%)	
Yes	159/269 (59%)	57/80 (71%)	
Unavailable	1/269 (0.4%)	0/80 (0%)	
Hemoglobin (g/dL)	12.18 (2.03)	11.67 (2.00)	0.047
Unknown	1	0	
Hemoglobin (g/dL)			0.127
Hb < 11	71/269 (26%)	30/80 (38%)	
Hb ≥ 11	197/269 (73%)	50/80 (63%)	
Unavailable	1/269 (0.4%)	0/80 (0%)	
Bone Marrow Involvement			0.293
No	243/269 (90%)	77/80 (96%)	
Yes	21/269 (7.8%)	3/80 (3.8%)	
Unavailable	5/269 (1.9%)	0/80 (0%)	
Extranodal Involvement			0.619
Extranodal sites = 0/1	199/269 (74%)	57/80 (71%)	
Extranodal sites ≥ 2	67/269 (25%)	23/80 (29%)	
Unavailable	3/269 (1.1%)	0/80 (0%)	
Ann Arbor Staging			0.007
Ann Stage I/II	115/269 (43%)	49/80 (61%)	
Ann Stage III/IV	153/269 (57%)	31/80 (39%)	
Unavailable	1/269 (0.4%)	0/80 (0%)	
Chemotherapy			0.004
CVP-R	2/269 (0.7%)	2/80 (2.5%)	
EPOCH-R	34/269 (13%)	17/80 (21%)	
De-Angelis-R	0/269 (0%)	2/80 (2.5%)	
CHOEP-R	1/269 (0.4%)	0/80 (0%)	
RCEPP	1/269 (0.4%)	0/80 (0%)	
MR-CHOP	17/269 (6.3%)	10/80 (13%)	
CHOP	3/269 (1.1%)	1/80 (1.3%)	
R-CHOP	193/269 (72%)	40/80 (50%)	
Others	18/269 (6.7%)	8/80 (10%)	

Abbreviations: CVP-R, Cyclophosphamide, Vincristine, Prednisolone, Rituximab; EPOCH-R, Etoposide, Prednisolone, Vincristine, Cyclophosphamide, Doxoubicin, Rituximab; De-Angelis-R, Rituximab, Methotrexate, Procarbazine, Vincristine; CHOEP-R, Cyclophosphamide, Doxoubicin, Vincristine, Etoposide, Prednisolone, Rituximab; RCEPP, Rituximab, Cyclophosphamide, Etoposide, Prednisolone, Procarbazine; MR-CHOP, Methotrexate, Rituximab, Cyclophosphamide, Doxoubicin, Vincristine, Prednisolone; CHOP, Cyclophosphamide, Doxoubicin, Vincristine, Prednisolone; R-CHOP, Rituximab, Cyclophosphamide, Doxoubicin, Vincristine, Prednisolone. ^1^ Data refer to mean (standard deviation). ^2^ Pearson’s Chi-squared test or Fisher’s exact test for categorical variable, and t-test for continuous variable.

**Table 2 cancers-16-03311-t002:** Univariate analysis of baseline characteristics associated with time-to-progression events.

		Events/Patients	HR (95% CI. *p-*Value)
Age	Per year increase	78/349	1.02 (1.00–1.04, *p* = 0.010)
Deauville Score	DV1-3	51/306	1.00 (ref)
	DV4-5	27/43	6.74 (4.21–10.79, *p* < 0.001)
International Prognostic Index	High ≥ 3	44/130	1.00 (ref)
	Low < 3	34/212	0.41 (0.26–0.64, *p* < 0.001)
	Unavailable	0/7	0.00 (0.00–Inf, *p* = 0.995)
Overall Bulky Status	bulky	13/48	1.00 (ref)
	not bulky	59/268	0.72 (0.39–1.31, *p* = 0.280)
	Unavailable	6/33	0.60 (0.23–1.58, *p* = 0.303)
ECOG Score	High ≥ 2	8/14	1.00 (ref)
	Low < 2	68/328	0.26 (0.13–0.55, *p* < 0.001)
	Unavailable	2/7	0.33 (0.07–1.59, *p* = 0.168)
Extranodal Involvement	No	26/132	1.00 (ref)
	Yes	52/216	1.41 (0.88–2.25, *p* = 0.157)
	Unavailable	0/1	0.00 (0.00–Inf, *p* = 0.996)
Presence of B Symptoms	No	32/185	1.00 (ref)
	Yes	33/105	2.10 (1.29–3.41, *p* = 0.003)
	Unavailable	13/59	1.32 (0.69–2.51, *p* = 0.404)
Receipt of Consolidative RT	RT-omitted	65/269	1.00 (ref)
	RT-treated	13/80	0.81 (0.45–1.48, *p* = 0.501)
Elevated LDH	No	12/83	1.00 (ref)
	Yes	65/260	1.89 (1.02–3.50, *p* = 0.043)
	Unavailable	1/6	1.23 (0.16–9.46, *p* = 0.843)
Hemoglobin Levels	Mean (SD)	78/348	0.84 (0.75–0.93, *p* = 0.001)
Marrow Involvement	No	64/320	1.00 (ref)
	Yes	13/24	2.76 (1.52–5.01, *p* = 0.001)
	Unavailable	1/5	2.06 (0.29–14.93, *p* = 0.473)
Ann Arbor Staging	Ann Stage I/IE	3/65	1.00 (ref)
	Ann Stage II/IIE	14/99	3.50 (1.01–12.18, *p* = 0.049)
	Ann Stage III/IIIE	22/52	13.06 (3.91–43.67, *p* < 0.001)
	Ann Stage IV	39/132	8.31 (2.57–26.89, *p* < 0.001)
	Unavailable	0/1	0.00 (0.00–Inf, *p* = 0.997)
Ann Arbor Staging	Ann Stage I/II	17/164	1.00 (ref)
	Ann Stage III/IV	61/184	3.94 (2.30–6.74, *p* < 0.001)
	Unavailable	0/1	0.00 (0.00–Inf, *p* = 0.997)

**Table 3 cancers-16-03311-t003:** Multivariable analysis of baseline characteristics associated with time-to-progression event.

Characteristic	HR	95% CI	*p*-Value
Receipt of Consolidative RT			0.009
RT-omitted	1.00 (ref)	—	
RT-treated	0.290	0.105, 0.801	
Deauville Score			0.000
DV4-5	1.00 (ref)	—	
DV1-3	0.108	0.060, 0.192	
Marrow Involvement			0.111
No	1.00 (ref)	—	
Yes	1.902	0.992, 3.646	
Unavailable	3.684	0.493, 27.52	
Hemoglobin Levels	0.967	0.851, 1.099	0.608
Presence of B Symptoms			0.044
No	1.00 (ref)	—	
Yes	1.797	1.037, 3.113	
Unavailable	0.844	0.427, 1.668	
International Prognostic Index			0.064
High ≥ 3	1.00 (ref)	—	
Low < 3	0.743	0.447, 1.236	
Unavailable	0.000	0.000, Inf	
Interaction term between Receipt of RT and Deauville Score	2.977	0.840, 10.55	0.087

**Table 4 cancers-16-03311-t004:** Multivariable interaction analysis.

	Unadjusted Model			Multivariable-Adjusted Model ^		
	HR (95% CI)	*p*	*p* (Int *)	HR (95% CI)	*p*	*p* (Int *)
TTP (78 events)						
Overall: RT-treated vs. RT-omitted	0.81 (0.45, 1.48)	0.492	-	0.80 (0.44, 1.47)	0.466	-
DV4-5: RT-treated vs. RT-omitted	0.33 (0.13, 0.88)	0.027	0.133	0.29 (0.10, 0.80)	0.017	0.087
DV1-3: RT-treated vs. RT-omitted	0.85 (0.40, 1.81)	0.671		0.86 (0.40, 1.86)	0.707	

^ Adjusted for International Prognostic Index, presence of B symptoms, bone marrow involvement, and hemoglobin levels. * Interaction term between RT and DV risk groups.

**Table 5 cancers-16-03311-t005:** Multivariable interaction analysis censored at 24 months.

	Unadjusted Model			Multivariable-Adjusted Model ^		
	HR (95% CI)	*p*	*p* (Int *)	HR (95% CI)	*p*	*p* (Int *)
TTP Censored at 24 months ^#^ (55 events)						
Overall: RT-treated vs. RT-omitted	0.83 (0.43, 1.60)	0.564	-	0.77 (0.40, 1.51)	0.437	-
DV4-5: RT-treated vs. RT-omitted	0.35 (0.13, 0.94)	0.037	0.166	0.28 (0.10, 0.79)	0.017	0.114
DV1-3: RT-treated vs. RT-omitted	0.90 (0.37, 2.20)	0.820		0.84 (0.34, 2.08)	0.711	

^ Adjusted for International Prognostic Index, presence of B symptoms, bone marrow involvement, and hemoglobin levels. * Interaction term between RT and DV risk groups. # Defined as the time from diagnosis to progression or relapse of DLBCL, or death due to DLBCL or treatment-related factors. If there was no event at the end of 24 months, patients were censored at the 24-month mark.

## Data Availability

Data used for analysis can be made available upon reasonable request to the corresponding author.

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
