# Peer review of "Using Deauville Scoring to Guide Consolidative Radiotherapy in Diffuse Large B-Cell Lymphoma"

_cancers, 2024, doi:10.3390/cancers16193311_

Round 1

Reviewer 1 Report

Comments and Suggestions for Authors

This study is a single institution study of DLBCL patient diagnosed 2010-2022 of the use of RT in patients completing 1st line immunochemotherapy:

There are some issues that must be clarified:

It is unclear how many cycles the patient received: Was it a inclusion criteria thar all cycles had been administered to the patient. Did some om the patient receive abbreviated treatment i.e. R-CHOP x 4 for limited disease?

Was dose reduction equal balanced between RT patients and non-RT patients?

Please clarify what MR-CHOP is, if M stands for Methrotrexate, it should be clear to the reader.

Was the PET/CT performed in accordance with the Lugano Criteria? It should be clarified when the EOT PET was performed.

Which strategy was used in order to decide which patients received RT, and who did not. Was it at the discretion of the clinician?

In Figure 1, several patients with high DS experience relapse 3-4 month after diagnose. I suspect that this is progressive disease at the EOT PET, and therefore they are excluded from RT, leading to a bias for this group?

There is increasing awareness that Deauville 4 is prognostic different to Deauville 5 in the response assessment EOT. Although the number in this study are low, the TTP curve for Deauville 4 and Deauville 5 should be show separately in a figure in the supplemental material. Deauville 1-3 can be omitted in this figure.

Reviewer 2 Report

Comments and Suggestions for Authors

Yau et al. report a retrospective single-centre analysis of the use of the PET-CT Deauville score at the end of treatment to assess whether radiotherapy is required after chemotherapy for diffuse large B-cell lymphoma. Patients with metabolic remission (Deauville 1-3) do better than patients with residual activitiy (Deavuville 4-5).

The authors compare two groups: All patients who radiotherapy and those who did not receive radiotherapy. The groups were balanced with respect to known DLBCL risk factors. In terms of treatment, there is a significant advantage for those patients with active residual disease who received radiotherapy, while patients with no or low activity did not benefit from additional therapy. Patients with active residual Deauville disease who did not receive radiotherapy suffer more recurrences. 

The radiotherapy is still relevant in multivariate risk factor analysis (IPI, Hb, Bone marrow involvement) 

Major: However from your manuscript i cannot figure out how you decided for or against radiotherapy please clarify

Limitations of retrospective single center analysis are numerous, however some additional date may help for future research.

Can you mention median and range for the time interval 

a) PET-CT relative to end of chemotherapy

b) radiotherapy relative to PET-CT

does this numbers have an effect on outcome , for example if very long intervals are harmful

Did you detect inter-observer differences in judgement of Deauville Score at your instution ? please comment

Comments on the Quality of English Language

Language only needs minor corrections by a native speaker. 
